# Evolution of the Probe-Based Loop-Mediated Isothermal Amplification (LAMP) Assays in Pathogen Detection

**DOI:** 10.3390/diagnostics13091530

**Published:** 2023-04-24

**Authors:** Xiaoling Zhang, Yongjuan Zhao, Yi Zeng, Chiyu Zhang

**Affiliations:** Shanghai Public Health Clinical Center, Fudan University, Shanghai 201508, China

**Keywords:** LAMP, probe, specificity, pathogen detection, sensitivity

## Abstract

Loop-mediated isothermal amplification (LAMP), as the rank one alternative to a polymerase chain reaction (PCR), has been widely applied in point-of-care testing (POCT) due to its rapid, simple, and cost-effective characteristics. However, it is difficult to achieve real-time monitoring and multiplex detection with the traditional LAMP method. In addition, these approaches that use turbidimetry, sequence-independent intercalating dyes, or pH-sensitive indicators to indirectly reflect amplification can result in false-positive results if non-specific amplification occurs. To fulfill the needs of specific target detection and one-pot multiplex detection, a variety of probe-based LAMP assays have been developed. This review focuses on the principles of these assays, summarizes their applications in pathogen detection, and discusses their features and advantages over the traditional LAMP methods.

## 1. Introduction

Pathogen detection is essential for the prevention and control of infectious diseases. The development of rapid, sensitive, specific, and cost-effective assays for the detection of pathogens (e.g., bacteria and viruses) remains a big challenge in the field of diagnosis [1,2,3]. This is largely due to the vast number and variety of pathogens, especially fast-mutating viruses [4].

Molecular diagnostics are commonly used in pathogen detection [5,6]. Currently, molecular diagnostic techniques can be broadly classified into the following categories: polymerase chain reaction (PCR), digital PCR, clustered regularly interspaced short palindromic repeats (CRISPR) test, isothermal acid amplification, biochip technique, and gene sequencing, etc. [5,6,7,8,9,10,11,12]. Since the revolutionary development of PCR in the 1980s, especially the development of quantitative (or real-time) PCR (qPCR) in the 1990s, nucleic acid amplification tests (NAATs) have become indispensable tools throughout the entire field of life science [13]. Currently, qPCR-based diagnostic procedures, with consistently reliable accuracy, have grown to be the gold standard for nucleic acid analysis and are widely used in laboratories and diagnostic facilities [7,14]. However, qPCR is not cost-effective because sophisticated instruments and technicians are required. In addition, qPCR is relatively time-consuming as it is cycled under strictly defined durations and temperature conditions [14,15]. Due to the limitations of qPCR mentioned above, a remarkable development trend of isothermal amplification methods has been observed since 1995. In recent years, isothermal NAATs have shown good performance in the rapid detection of pathogens [16]. This technology can be simply performed in low-cost devices and has a fast processing time compared to qPCR, making it particularly suitable for point-of-care testing (POCT) [17,18,19]. After the pioneering development of the rolling circle amplification (RCA), various alternative approaches were developed to enable isothermal amplification, mainly including loop-mediated isothermal amplification (LAMP), nucleic acid sequence-based amplification (NASBA), recombinase polymerase amplification (RPA), helicase-dependent amplification (HDA), and so forth [20,21,22,23,24,25,26]. LAMP is one of the most frequently used isothermal amplification methods [16].

LAMP was first described by Notomi et al. in 2000 [23]. It utilizes two pairs of primers, including two inner primers (forward: FIP (F1c + F2) and backward: BIP (B1c + B2)) and two outer primers (F3 and B3), to initiate isothermal DNA amplification at a stable temperature of 60–65 °C by *Bst* DNA polymerase with a strand displacement activity (Figure 1). The inner primers contain two target sequences specific to two different regions in plus and minus strands of the template DNA. LAMP amplification contains two phases, an initial amplification phase, and an exponential amplification phase. In the initial amplification phase, the inner primers hybridize with the target DNA and start complementary strand synthesis (Figure 1A). At the same time, the outer primers also initiate DNA synthesis to displace the newly synthesized DNA strands initiated by the inner primers (e.g., FIP) (Figure 1B). The released single-stranded (ss) DNA further serves as a template for DNA synthesis with the corresponding inner primers (e.g., BIP) (Figure 1C) and outer primers (e.g., B3) (Figure 1D). The newly released ssDNA can fold into a dumbbell DNA molecule not only to initiate the self-primed DNA synthesis but also to serve as a template for new amplification cycles using the inner primers (Figure 1E–G).

In 2002, Nagamine et al. further optimized the LAMP method to accelerate amplification by adding additional loop primers [27]. Loop primers (forward: LF and backward: LB) bind to the loop regions of the complex dumbbell structure and promote new DNA synthesis to accelerate isothermal amplification (Figure 1H). As a result, the number of LAMP products can increase even up to a billion copies in less than an hour [23,27]. LAMP has been demonstrated to have high sensitivity and good tolerance to PCR-inhibiting substances [15,16]. LAMP also exhibits the versatility of result reading and the ability to expand detection in combination with other technologies. Among the methods most widely used for the detection of LAMP signal are turbidity caused by precipitated magnesium pyrophosphate, sequence-independent intercalating dyes like metal indicators for calcium, intercalating fluorescent dyes such as SYBR green and STYO 9, and pH-sensitive indicators (e.g., Cresol red) [15,28,29]. In addition, LAMP can be combined with other different technologies to further expand its detection capacity. For example, magnetic beads or gold nanoparticles (AuNPs) can be used to create easily manageable lateral flow tests in combination with LAMP assays [30,31]. Furthermore, LAMP amplicon detection can also be combined with various sequencing techniques, such as nanopore sequencing [32], pyrosequencing [33], or high-throughput sequencing [34], to realize multiplex detection of different pathogens (e.g., Severe Acute Respiratory Syndrome Coronavirus 2 (SARS-CoV-2) and other respiratory viruses).

Despite the advantage in sensitivity and amplification speed, the traditional LAMP methods have some limitations. The major limitation is that LAMP frequently yields spurious amplicons if the primers interact with each other [35], which can result in false positive results when sequence-independent intercalating dyes or pH-sensitive indicators are used [35,36,37,38]. Another limitation is that it is difficult for LAMP to achieve single-tube multiplex detection, and as a consequence, most traditional LAMP methods only detect one target in a single reaction [15,35,39]. To fulfill the needs of specific target detection and one-pot multiplex reaction, several probe-based LAMP assays have been developed (Figure 2). These probe-based LAMP assays not only guarantee the accurate detection of the target without being affected by non-specific products but also enable the multiplex detection of various targets in a single reaction [16].

Herein, we review the evolution of probe-based LAMP assays, describe the principles of these methods, and summarize their applications in pathogen detection. Furthermore, the review discusses the advantages of probe-based LAMP assays over the traditional LAMP methods and highlights their superiority and flexibility in clinical application.

## 2. The Principles of the Probe-Based LAMP Assays

In the probe-based LAMP assays, different probes were designed for LAMP assays, including the modification of one of the core LAMP primers and/or the introduction of additional probes or additives to the reaction [16,40].

### 2.1. Assimilating Probe-Based LAMP Assay

In 2011, Kubota et al. developed the assimilating probe-based LAMP assay [41]. The assimilating probes consist of two partially complementary oligonucleotides: a fluorescence probe and a quenching probe. The fluorescence probe was designed by adding a fluorophore (denoted by F, red)-labeled universal oligonucleotide (F strand) to the 5′-end of the loop primer (LF or LB). The quenching probe (Q strand) complements the F strand of the fluorescence probe and is labeled with a quencher (denoted by Q, blue) at its 3′ end. In the absence of target DNA, the fluorescence is quenched since the hybridization of the fluorescent probe and the quenching probe brings the quencher and the fluorophore together (Figure 3A). In the presence of target DNA, LAMP is initiated. Along the DNA strands initiated by the fluorescent probe, the newly synthesized DNA strand displaces the quenching probe to release the fluorescent signal of the fluorophore (Figure 3B).

### 2.2. Detection of Amplification Using Release of Quenching (DARQ)

The DARQ was first reported by Tanner in 2012 [42]. Similar to the assimilating probe-based LAMP assay, a partially double-stranded linear DNA probe was also used in DARQ. The probe was designed by labeling a quencher at the 5′ end of the FIP and adding an additional 3′ fluorophore-labeled probe that is complementary to F1c of FIP (Figure 4A). In the absence of target DNA, no fluorescent signal is generated from the double-stranded linear DNA probe. During the LAMP amplification, the newly synthesized DNA strand displaces the 3′ fluorophore-labeled probe to release a fluorescent signal (Figure 4B).

### 2.3. Quenching Probe (Q Probe)-Based LAMP Assay

In 2018, Shiratoa et al. developed a reverse transcription loop-mediated isothermal amplification (RT-LAMP) assay using a quenching probe (Q probe) [43]. The Q probe that targets the loop regions of the LAMP amplicons was simply designed by lengthening the 3′ end of one loop primer by 10–15 nt and featured by a 3′ fluorophore (e.g., BODIPY)-labeled cytosine (Figure 5A). When the Q probe hybridizes with the target sequence, the fluorescence is quenched by the guanine residue present in the target sequences (Figure 5B). In 2019, Takayama et al. further simplified the design of the Q probe by directly labeling the fluorophore at the 5′ cytosine of one of the loop primers [44]. Distinct from the double-stranded linear DNA probes described above, the fluorescent signal of the Q probe is released in the absence of target DNA, and the fluorescence is quenched when the Q probe hybridizes with the targets. As a result, the fluorescent signal decreases, accompanied by the accumulation of amplicons during LAMP amplification. The LAMP amplicons can be real-time monitored using a real-time fluorescent PCR instrument, and exponential amplification results in an inverted S curve with continuous quenched fluorescent signals.

### 2.4. Enzyme-Mediated Probe-Based LAMP Assays

Enzyme-mediated probe-based LAMP assays utilize an extra enzyme (e.g., endonuclease, high fidelity DNA polymerase) or *Bst 5.0/5.1 DNA polymerase* to recognize and cut the probes to release fluorescent signals.

#### 2.4.1. Multiple Endonuclease Restriction Real-Time Loop-Mediated Isothermal Amplification (MERT-LAMP)

Wang et al. developed MERT-LAMP in 2015 [45]. In MERT-LAMP, the probe was designed by linking an additional oligonucleotide to the 5′ end of an inner primer (e.g., FIP), and the oligonucleotide carried a 5′ fluorophore and a 3′ quencher, both of which were separated by a restriction endonuclease recognition site (Figure 6A). In the initial amplification phase, the forward inner primer FIP with endonuclease recognition site binds to the F2c region to promote the synthesis of DNA. The newly synthesized strand is displaced by upstream DNA synthesis primed using the outer primer F3. The BIP annealing to the B2c sequence synthesizes the complementary sequence of the oligonucleotide probe. When the double-stranded DNA containing the endonuclease recognition site is generated, *Nb.BsrDI* recognizes the restriction endonuclease site and digests the double-stranded DNA. This process separates the fluorophore and the quencher, resulting in a fluorescent signal (Figure 6B). Accompanied by the exponential amplification of LAMP, fluorescence signals are continuously generated.

#### 2.4.2. Tth Endonuclease Cleavage Loop-Mediated Isothermal Amplification (TEC-LAMP)

Higgins et al. developed TEC-LAMP in 2018 [46]. Instead of an endonuclease recognition site, an abasic site that can be cleaved with *Tth endonuclease IV* was used to link the fluorophore and the quencher. The reaction process is similar to that of MERT-LAMP (Figure 6).

#### 2.4.3. High-Fidelity DNA Polymerase-Mediated LAMP

To overcome the shortcomings of existing LAMP methods for detecting SARS-CoV-2, three high-fidelity DNA polymerase-mediated real-time LAMP methods with similar principles have been developed since 2021 [36,37,38]. The release of a fluorescent signal was termed “proofreading” enzyme-mediated probe cleavage (Proofman probe) [36], high-fidelity DNA polymerase mediated probe (HFman probe) [37], and Luminescence from ANticipated Target due to Exonuclease Removal of Nucleotide mismatch (LANTERN) [38]. In these assays, one of the loop primers with or without a 3′-mismatched base is selected as the probe by labeling a fluorophore and a quencher at the 3′ end and 5′ end, or the 5′ end and 3′ end, respectively (Figure 7A). To perform real-time monitoring, a small amount of high-fidelity DNA polymerase is introduced into the LAMP system. During LAMP amplification, the probes specifically bind to the loop regions of the dumbbell structure, and then high-fidelity DNA polymerase recognizes and cuts 3′-matched and/or mismatched base labeled with a fluorophore to release fluorescent signal (Figure 7B) [36,37,38]. Previous studies showed that high-fidelity DNA polymerase could well recognize and cut the probes that specifically bind to the target sequence, regardless of the presence or absence of a mismatched base at the 3′ end of the probes or primers [37,47,48]. Furthermore, after removing the 3′-labeled base, the probe exposes a 3′-OH and then is used as a primer to initiate new DNA synthesis using *Bst DNA polymerase*, which facilitates the amplification.

#### 2.4.4. LAMP Coupled with TaqMan Probe and a New Generation *Bst* DNA Polymerase (Taqman-LAMP)

In 2021, Liang et al. reported the development of a TaqMan-LAMP, which used an innovative *Bst 5.0/5.1 DNA polymerase* (Figure 8A) [49]. Differing from *Bst 4.0 DNA polymerase* and earlier *Bst DNA polymerases*, the innovative *Bst 5.0/5.1 DNA polymerase* does not only have a DNA synthesis activity and a stronger strand displacement capacity but also a 5′-3′ exonuclease activity. The sequence located between F1 and F2 is selected as a target for the design of a TaqMan probe that is labeled with fluorophore and quencher at its 5′ end and 3′ end, respectively. The inner primer FIP is annealed to F2c to promote DNA synthesis. When the Taqman probe specifically binds to the target, *Bst 5.0/5.1 DNA polymerase* can cleave the TaqMan probe to release the fluorescent signal (Figure 8B). During the exponential amplification phase, ongoing binding and cleavage of the TaqMan probes result in a continuous generation of fluorescent signals.

## 3. Applications of the Probe-Based LAMP Methods in Pathogen Detection

The urgent demand for specific and accurate POCT detection of emerging and reemerging infectious diseases has promoted the development of various probe-based LAMP methods [17,18,19,50]. The applications of various probe-based LAMP assays in pathogen detection are summarized in Table 1.

Kubota et al. developed the assimilating probe-based LAMP and subsequently designed a simple, low-power, portable handheld device to enable real-time detection of the LAMP amplicons [41,51]. This assay was mainly used for singleplex detection of viruses, and the sensitivities of this assay in the detection of foot-and-mouth disease virus (FMDV) and porcine circovirus 3 (PCV3) were 10^2^ copies per µL and 50 copies per reaction, respectively [52,53].

The DARQ was first reported by Tanner et al. for singleplex and quadruplex detection of *E. coli, Caenorhabditis elegans*, the human cystic fibrosis transmembrane conductance regulator, and human BRCA1 DNA [42]. Since its development, the assay has been applied in singleplex or multiplex detection of other pathogens [54,55,56,57,58,59,60]. For example, Fan et al. used DARQ to develop a visual multiplex fluorescent LAMP assay for the detection of FMDV, vesicular stomatitis virus (VSV), and bluetongue virus (BTV) [59]. The detection limit was as low as 526–2477 copies/reaction for plasmid standards of these viruses, indicating that the multiplex LAMP assay can simultaneously detect 1–3 targets in a single reaction tube. The same team further developed a multiplex fluorescence-based LAMP assay for detecting chicken parvovirus (ChPV), chicken infectious anemia virus (CIAV), and fowl adenovirus serotype 4 (FAdV-4), with detection limits of 307, 749 and 648 copies per reaction, respectively [60].

Takayama et al. developed a reverse transcription loop-mediated isothermal amplification (RT-LAMP) assay using a Q probe for the detection of the influenza virus (IV), respiratory syncytial virus (RSV), and rhinovirus [44,61]. Shirato et al. used the Q probe to develop a real-time LAMP assay for the detection of the Middle East respiratory syndrome coronavirus (MERS-CoV) [43]. The Q probe-based LAMP was also applied to detect Genus *Phytophthora*, Species *Phytophthora,* and severe fever with thrombocytopenia syndrome virus (SFTSV) [62,63]. However, all the above-mentioned Q probe-based LAMP assays were developed in a singleplex format.

In order to achieve single-tube multiplex detection of *Listeria monocytogenes* and *Listeria ivanovii*, Wang et al. developed MERT-LAMP [45]. As low as 250 fg of DNA per reaction of tested *Listeria* DNA could be detected. Higgins et al. developed TEC-LAMP for the simultaneous detection of *Streptococcus pneumoniae*, *Neisseria meningitidis*, and *Haemophilus influenzae* [46]. The limit of detection (LOD) of the LEC-LAMP assay was 3.1 genome copies per reaction for the singleplex detection of *N. meningitidis* and 10^2^ genome copies per reaction for multiplex detection of all three bacteria [46].

In the high-fidelity DNA polymerase-mediated LAMP assay, Ding et al. first introduced a thermostable proofreading DNA polymerase (Pfu) to trigger the cleavage of the Proofman probe [36]. The N and Orf1ab genes of SARS-CoV-2 and human glyceraldehyde 3-phosphate dehydrogenase (GAPDH) gene were simultaneously detected with different fluorophores, and the detection limit was reported to be 100 SARS-CoV-2 RNA copies per reaction [36]. However, the performance of the SARS-CoV-2 RT-LAMP assay was not evaluated using clinical samples. Soon, a more sensitive SARS-CoV-2 RT-HF-LAMP assay was developed using an HFman probe [37]. Subsequently, HFman probe-based LAMP (HF-LAMP) was used to develop various assays for singleplex or multiplex detection of human immunodeficiency virus-1 (HIV-1), Hantaan virus (HTNV) and Seoul virus (SEOV), Monkeypox virus (MPXV) and Monkey B virus (BV), and BK virus (BKV), and the detection sensitivity can achieve three copies of viral RNA per reaction [64,65,66,67]. Evaluation with clinical samples showed an excellent consistency of HF-LAMP assays to commercial RT-qPCR assays. Importantly, the HF-LAMP method was demonstrated to have high detection potential for highly variable viruses and can be developed to an extraction-free format for the detection of various human viruses directly using clinical samples [64,65,66,67].

Liang et al. developed a TaqMan-LAMP and used this method for rapid and specific detection of the F gene of Pigeon paramyxovirus-1 (PPMV-1) [49]. The assay could be completed within 25 min, and the LOD was estimated to be 10 copies per µL for PPMV-1 cDNA and 0.1 ng for PPMV-1 RNA [49].

More application examples of these probe-based LAMP methods in the detection of various pathogens can be found in Table 1.

**Table 1 diagnostics-13-01530-t001:** Applications of the probe-based LAMP methods in pathogen detection.

Methods	Year	Singleplex or Multiplex	Targets	Reaction Condition	Sensitivity	Limitation of Detection	Clinical Evaluation	Tt Values	Ref.
Assimilating probe	2011	singleplex	*Ralstonia solanacearum*	65 °C for 60 min	50 fg	137 copies per reaction	NE *	consistently around 20 min regardless of copy number (10^2^ to 10^6^ gene copies)	[41]
Assimilating probe	2011	singleplex	*Salmonella enterica*	65 °C for 60 min	5 pg	15 copies per reaction	NE	NA **	[51]
Assimilating probe	2015	multiplex	*Salmonela enterica* and Phage λ	65 °C for 30 min	50 fg	9.8 × 103 and 1000 copies per reaction for Salmonella enterica and Phage λ	NE	NA	[68]
Assimilating probe	2020	singleplex	foot-and-mouth disease virus	62 °C for 30 min	100 copies	2.5 × 103 copies per reaction	69 clinical samples (28 blood, 28 oropharyngeal fluid, and 13 tissues)	4.19–15.98 (9.95 ± 3.08) min	[52]
Assimilating probe	2020	singleplex	Porcine circovirus type 3	62 °C for 30 min	50 copies	NA	326 pig samples (136 tissues and 190 sera)	8.42–27.73 (17.34 ± 4.45) min	[53]
DARQ	2012	multiplex	Bacteriophage λ, HeLa, *Escherichia coli*, and *Caenorhabditis elegans* genomic DNA	65 °C for 60 min	5 ng, 10 pg, 5 ng, 82.5 ng for Bacteriophage λ, HeLa, Escherichia coli, and Caenorhabditis elegans genomic DNA, respectively.	NA	NE	11.8 ± 0.03 min	[42]
DARQ	2016	singleplex	*Salmonella*	65 °C for 40 min and inactivated at 85 °C for 5 min.	10 copies	NA	312 fecal samples	NA	[54]
DARQ	2017	singleplex	avian reovirus	65 °C for 60 min	10 copies	NA	98 clinical tendon tissue samples	NA	[55]
DARQ	2019	multiplex	methicillin-resistant *Staphylococcus aureus* (MRSA)	65 °C for 60 min	103 copies, 103 copies, 104 copies for singleplex detection of femB, spa, and mecA of MRSA genes.104 copies for duplex and triplex detection.	NA	NE	NA	[56]
DARQ	2019	singleplex	*Brucella*	65 °C for 45 min and terminated at 85 °C for 5 min.	20 copies	NA	250 samples	NA	[57]
DARQ	2021	multiplex	SARS-CoV-2, influenza A, influenza B, human RNA	60 °C, 15 s for 108 cycles	50 copies of SARS-CoV-2 RNA, 1:10000 diluted influenza A RNA (VR-1737D) and approximately 21 copies of influenza B RNA (VR-1885DQ).	NA	NE	NA	[58]
DARQ	2022	multiplex	foot-and-mouth disease, vesicular stomatitis, and bluetongue viruses	63 °C for 75 min, 80 °C for 5 min for termination	1000 copies for FMDV, 100 copies for VSV, BTV-4	2477copies/reaction for FMDV, 526 copies/reaction for VSV, and 913 copies/reaction for BTV	111 clinical samples, including 12 vesicular fluid samples, 30 esophageal–pharyngeal samples, 42 whole-blood samples, 6 vesicular skin samples, and 21 oral swabs collected from cattle	NA	[59]
DARQ	2023	multiplex	Chicken parvovirus (ChPV), chicken infectious anemia virus (CIAV), and fowl adenovirus serotype 4 (FAdV-4)	63 °C for 75 min for amplification and 80 °C for 5 min for termination	66,106,95 copies for ChPV, CIAV, and FAdV-4, respectively.	307 copies/µL for ChPV, 749 copies/µL for CIAV and 648 copies/µL for FAdV-4	342 samples (cloacal swab, heart, liver, and kidney) collected from chicken farms	NA	[60]
Q Probe	2017	singleplex	*Fusarium oxysporum f. sp. Lycopersici*	66 °C for 60 min, and subsequent melting curve analysis from 30 or 35 °C to 95 °C with a decrement of 0.2 °C per second.	3 ng	NA	4 Soil DNA	NA	[69]
Q Probe	2018	singleplex	Middle East respiratory syndrome coronavirus	63 °C for 30 min	20 copies	NA	19 nasal aspirates, secretions, or swabs	20.2–30.9 min	[43]
QProbe	2020	singleplex	Genus *Phytophthora* and Species *Phytophthora*	preheat 5 min at 68 °C, 68 °C for 60 min, and annealing curve analysis at 98 °C to 80 °C, ramping at 0.05 °C per sec.	100 fg	NA	161 taxa, including subspecies, varieties, and hybrids) of phytophthora, 12 species (12 isolates) of Pythium, 17 species (17 isolates) of Phytopythium, and one isolate of each of the soil-borne pathogens.	NA	[62]
Q Probe	2021	singleplex	severe fever with thrombocytopenia syndrome virus (SFTSV)	63 °C for 30 min	10–100 copies	NA	12 SFTSV strains	NA	[63]
Enzyme-mediated probe-based LAMP assays (MERT-LAMP)	2015	multiplex	*L. monocytogenes* and *L. ivanovii*	64 °C for 60 min and then at 80 °C for 5 min	250 fg	NA	70 raw meat samples (including pork, beef, lamb, and chicken samples)	NA	[45]
Enzyme-mediated probe-based LAMP assays (TEC-LAMP)	2018	multiplex	*Streptococcus pneumoniae, Neisseria meningitidis, Haemophilus influenzae*, and Internal Amplification Control	67 °C for 60 min	100 copies for S. pneumoniae, N. meningitidis, and H. influenzae, 50 copies for Internal Amplification Control	39.5, 17.3, and 25.9 genome copies per reaction for S. pneumoniae, N. meningitidis, and H. influenzae, respectively.	65 samples, including 34 blood, 5 blood culture, 17 CSF, 5 pleural fluid, 1 knee fluid, and 3 other body fluids	NA	[46]
Enzyme-mediated probe-based LAMP assays (Proofman)	2021	multiplex	N and Orf1ab genes SARS-CoV-2, human glyceraldehyde-3-phosphate dehydrogenase (GAPDH)	60 °C for 60 min	100 copies of N gene RNA	100 copies per reaction	NE	NA	[36]
Enzyme-mediated probe-based LAMP assays (HFman)	2022	multiplex	ORF and E genes of SARS-CoV-2, human β-actin gene	64 °C for 50 min	30 copies of ORF and E gene RNA	78 and 115 copies per reaction for ORF gene and E gene, respectively.	190 nasopharyngeal swabs (NP) samples	4.0–33.5 min (ORF gene),4.0–45.0 min (E gene)	[37]
Enzyme-mediated probe-based LAMP assays (HFman)	2022	singleplex	HIV-1 subtypes, including CRF01_AE, CRF07_BC, CRF08_BC, CRF55_01B, and unique recombinant forms (URFs).	64 °C for 50 min	3 copies	89 copies per reaction	101 plasma sample	9.67–29.98 min	[64]
Enzyme-mediated probe-based LAMP assays (HFman)	2022	multiplex	Hantaan virus (HTNV) and Seoul virus (SEOV)	64 °C for 50 min	3 copies	41 and 73 copies per reaction for HTNV and SEOV, respectively.	46 serum samples	NA	[65]
Enzyme-mediated probe-based LAMP assays (HFman)	2023	multiplex	Monkeypox virus (MPXV) and Monkey B virus (BV)	64 °C for 50 min	3 copies	28.7 and 27.8 copies per reaction for MPXV and BV, respectively.	simulated 6 serum samples collected from monkeys	NA	[66]
Enzyme-mediated probe-based LAMP assays (HFman)	2023	singleplex	BK virus (BKV)	64 °C for 50 min	3 copies	12 copies/reaction	132 urine samples from HIV-1 infected individuals, 20 BKV positive samples, and 10 BKV negative samples	3.9 to 15.4 min	[67]
Enzyme-mediated probe-based LAMP assays (LANTERN)	2022	multiplex	S gene of SARS-CoV-2 and human ACTB	65 °C for 40 min	2 copies	8 copies per reaction	52 COVID-19 positive samples and 22 COVID-19 negative samples	NA	[38]
Enzyme-mediated probe-based LAMP assays (TaqMan-LAMP)	2021	singleplex	pigeon paramyxovirus type 1	65 °C for 10 s and 65 °C for 50 s with 25 cycles.	10 copies	NA	108 fecal samples	NA	[49]

* NE, not evaluated. ** NA, not available.

## 4. Comparison of the Probe-Based LAMP Methods with the Traditional LAMP Methods

The probe-based LAMP methods were developed from the traditional LAMP methods. Because of the use of a probe in the reaction system, the instruments and detection system for the probe-based LAMP methods are different from those for the traditional LAMP methods (Table 2). Real-time thermal cycler or small portable isothermal fluorimeters are required for the real-time monitoring of LAMP amplicons, or regular thermal cycler or low-cost simple heating block (e.g., dry incubator) for end-point fluorescence detection of the products. Therefore, the facility for the probe-based LAMP methods is more likely to be more costly than the traditional LAMP methods. It is worth noting that some portable fluorescence analyzers and isothermal fluorimeters with minimal laboratory resources are currently available or under development (Appendix A) [70,71,72,73,74,75]. Furthermore, end-point fluorescence detection requires much less instrumentation than real-time monitoring. Using a simple, low-cost device, such as a UV transilluminator [76] or light-boxes using cardboard and filter paper [38], the results can be easily achieved post-amplification.

Since the first description of the LAMP method in 2000, a few LAMP assays have been commercially used [77]. The major reason is the low specificity and accuracy of the traditional LAMP methods, which are largely ascribed to the inevitable introduction of non-specific amplification during the reaction and the use of sequence-independent double-stranded DNA binding dyes or pH-sensitive indicators [15,77]. The use of probes can largely improve the specificity of NAATs since the specific hybridization of a probe with its target is a prerequisite for enzymes to recognize and cut the probe and further release fluorescence signals [16]. Therefore, like probe-based qPCR assays, the probe-based LAMP assays are highly specific and accurate for the detection of various pathogens compared to the traditional LAMP assays, and meanwhile, the use of different probes confers the single tube multiplex detection capacity of the probe-based LAMP assays (Table 2) [16]. Furthermore, the sensitivities of the probe-based LAMP methods are slightly higher, and the process is slightly easier than the traditional LAMP methods.

## 5. Comparison of Various Probe-Based LAMP Methods

The various probe-based LAMP methods exhibit distinct features due to being derived from different principles (Table 3). Of these methods, the probe designs for assimilating probe-LAMP, Q probe-LAMP, MERT-LAMP/TEC-LAMP, and Taqman-LAMP are relatively difficult. The Assimilating probe-LAMP requires an additional oligonucleotide probe, and the TaqMan-LAMP requires the design of a probe between F1 and F2 or B1 and B2. Furthermore, a terminal cytosine and an endonuclease recognition site are required for the probe of the Q probe-LAMP and the MERT-LAMP/TEC-LAMP, respectively. Although the probe-based LAMP methods largely improve the specificity of the assays, the specificity of the Assimilating probe-LAMP, DARQ, and MERT-LAMP/TEC-LAMP are relatively lower compared to the other three methods. The main reason is that their probes are involved in the inner primers that more likely yield non-specific amplification and the release of the fluorescent signal depends on the strand displacement of the double-stranded linear probes.

According to the principles, all these probe-based methods have the capacity for multiplex detection. However, it is relatively difficult for these LAMP methods to realize single-tube high-sensitive detection of three or more targets (triple or quadruple) since the presence of too many primers might cause complex primer–primer interaction and therefore reduce amplification efficiency (detection sensitivity) (Table 1 and Table 3). The MERT-LAMP/TEC-LAMP and the Proofman/HFman/LANTERN-based methods appear to have higher detection sensitivity than other probe-based methods. All these methods have faster amplification than the traditional LAMP assays, and most of these methods have Tt values of less than 30 min.

On the other hand, most emerging and remerging infectious diseases are caused by viruses (especially RNA viruses such as the recent SARS-CoV-2 and influenza viruses), which are more variable. The presence of mismatched between primers/probes and viral genomic sequence may reduce LAMP amplification efficiency, resulting in lower detection sensitivity and even a failure in detection [28]. The introduction of high-fidelity DNA polymerase enables Proofman/HFman/LANTERN-based LAMP assays with high potential to detect various viral variants (Table 3) [28,30,37,78].

## 6. Conclusions and Future Perspectives

As a promising method for nucleic acid detection, LAMP has attracted more and more attention in the last two decades for POCT detection of animal and plant viruses, bacteria, fungi, mycoplasma, parasites, and other pathogens due to its high sensitivity, rapidity, and simplicity [6,19,26,79,80]. The major reason making the traditional LAMP methods inferior to qPCR assays and limiting their commercial and clinical application is their low specificity and accuracy, which are ascribed to non-specific amplification and/or the use of sequence-independent interferents (e.g., non-specific double-stranded DNA binding dyes) [15,35,77]. The developments of various probe-based LAMP methods not only resolve the problems of non-specific detection and low accuracy but also improve the amplification speed and detection sensitivity and enable single-pot multiplex detection [37,38,40]. In view of its advantages over the traditional LAMP methods and its comparability to qPCR in specificity, sensitivity, multiplex detection capacity, and faster speed than the latter, the probe-based LAMP methods show high commercial and clinical application potential to be readily used, especially in the detection of emerging and reemerging infectious diseases.

Despite the promising advancements of the probe-based LAMP methods, two underlying questions remain to be further addressed for future POCT applications at home and in the field. First, purified nucleic acids are often used in various NAATs. Although several extraction-free probe-based LAMP assays were developed for SARS-CoV-2 detection [81,82,83,84], the extraction-free assays largely depend on the sample types. For example, because of the relatively low inhibition of nasopharyngeal swabs (NPs) on amplification reactions, NPs are more suitable for extraction-free nucleic acids detection regardless of qPCR and LAMP than other sample types (e.g., plasma/serum and tissues) [67,84]. Therefore, additives that can mitigate the inhibition of clinical samples on LAMP amplification deserve to be developed. Furthermore, a suitable buffer system for the collection of NPs and other samples also needs to be developed since currently used commercial viral transport media often contain substances inhibiting LAMP amplification [37,67].

Second, the results of the probe-based LAMP methods can be real-time monitored using a real-time (thermal or isothermal) fluorimeter or be observed at the end-point using a regular fluorimeter [70,71,73]. A small, portable, cheap, and battery-powered isothermal fluorimeter should be developed. In particular, a small portable device that integrates rapid nucleic acid extraction, isothermal amplification, and fluorescence monitoring is strongly encouraged.

## Figures and Tables

**Figure 1 diagnostics-13-01530-f001:**
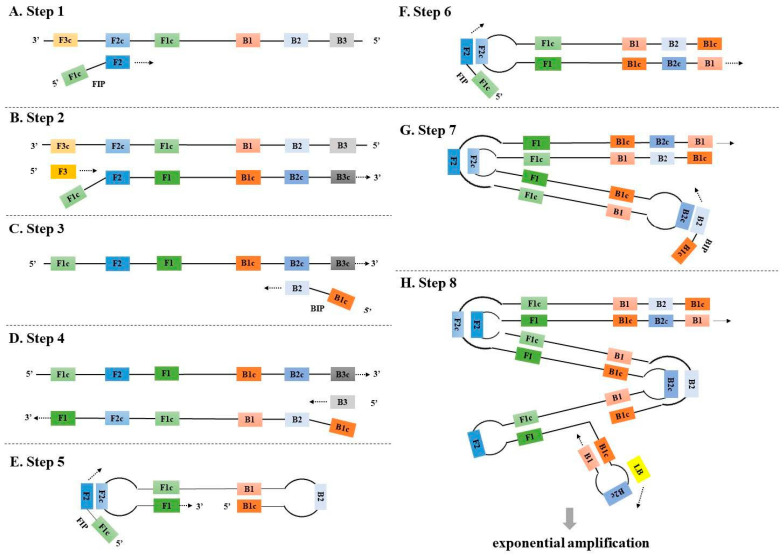
The principle of the traditional LAMP. F3 and B3: Forward and backward outer primers, respectively; FIP (F1c + F2) and BIP (B1c + B2): Forward and backward inner primers, respectively; LF and LB: Forward and backward loop primers, respectively.

**Figure 2 diagnostics-13-01530-f002:**
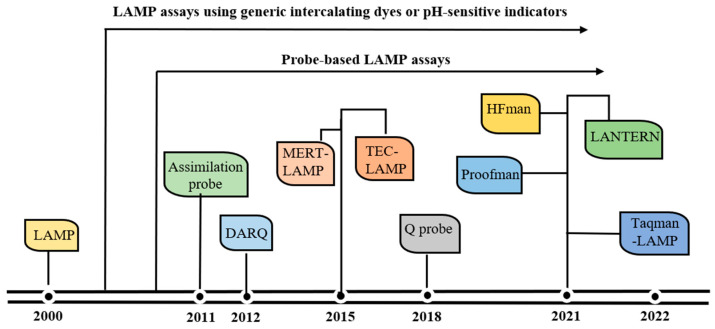
Evolution of probe-based LAMP assays.

**Figure 3 diagnostics-13-01530-f003:**
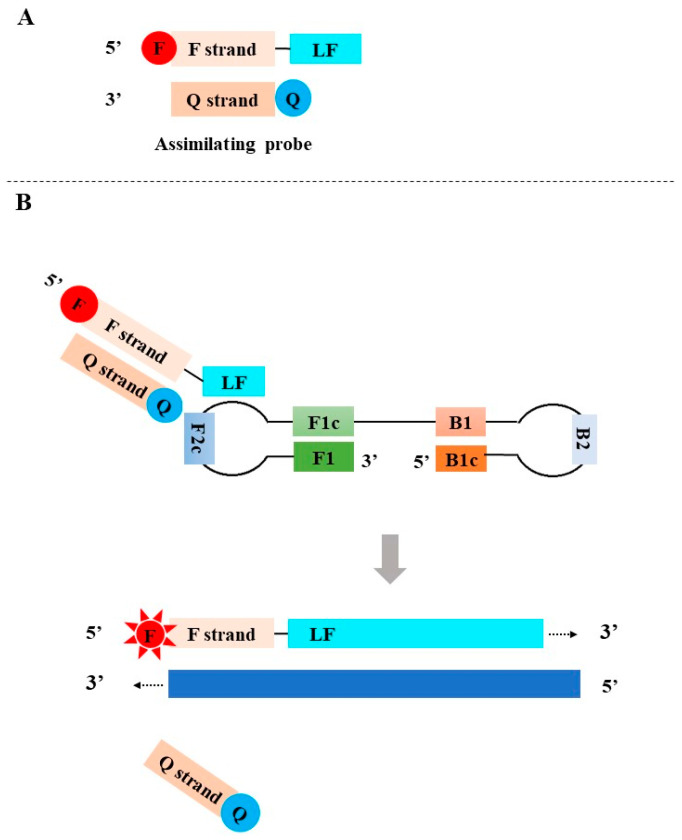
The principle of the assimilating probe-based LAMP assay. F: fluorophore; Q: quencher.

**Figure 4 diagnostics-13-01530-f004:**
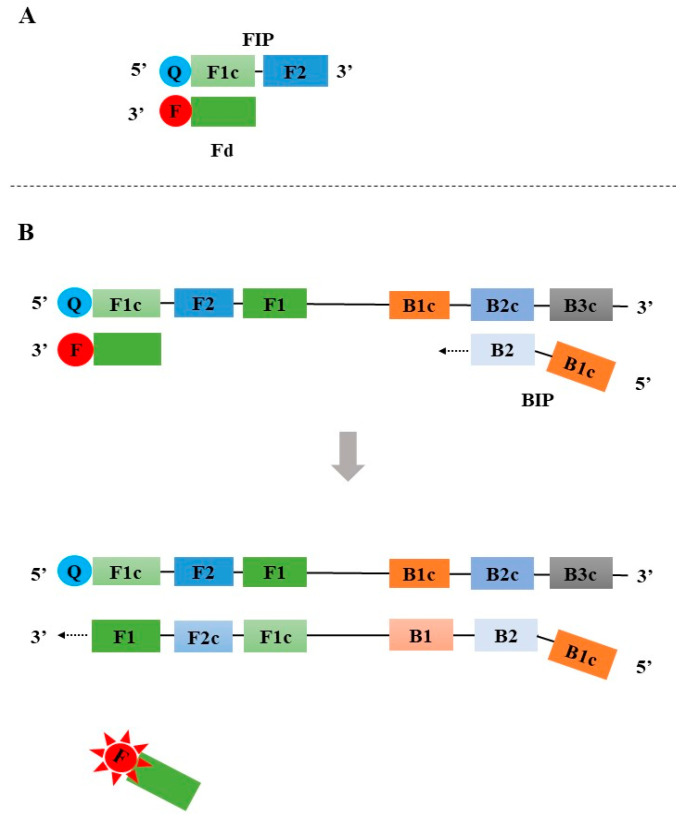
The principle of the Detection of amplification using the release of quenching (DARQ). F: fluorophore; Q: quencher.

**Figure 5 diagnostics-13-01530-f005:**
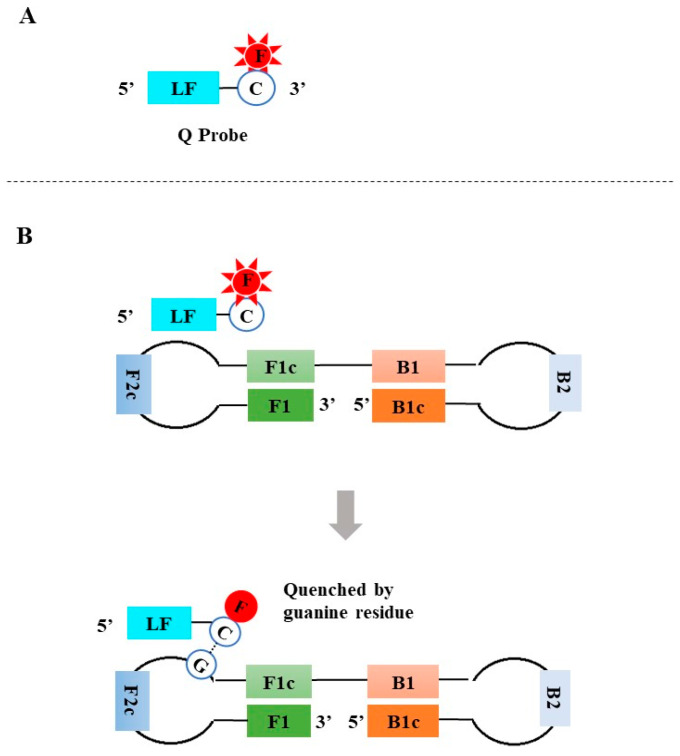
The principle of the Quenching probe (Q probe)-based LAMP assay. F: fluorophore; Q: quencher; C: cytosine.

**Figure 6 diagnostics-13-01530-f006:**
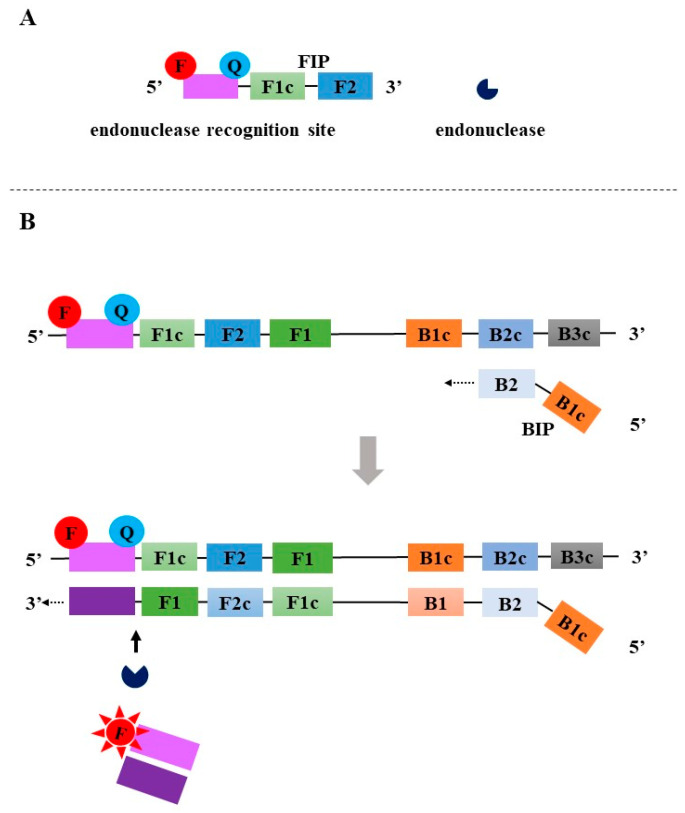
The principle of the MERT-LAMP. F: fluorophore; Q: quencher.

**Figure 7 diagnostics-13-01530-f007:**
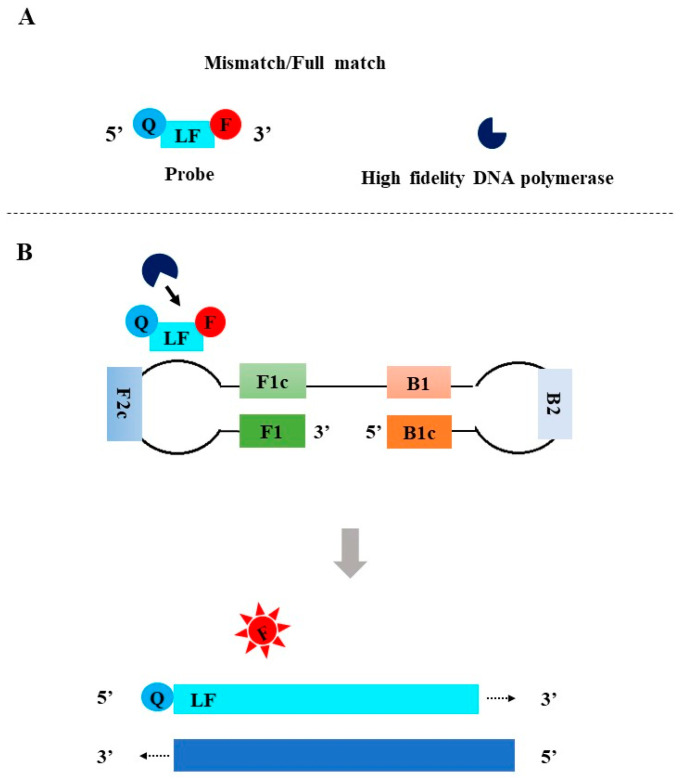
The principle of High-fidelity DNA polymerase-mediated LAMP. F: fluorophore; Q: quencher.

**Figure 8 diagnostics-13-01530-f008:**
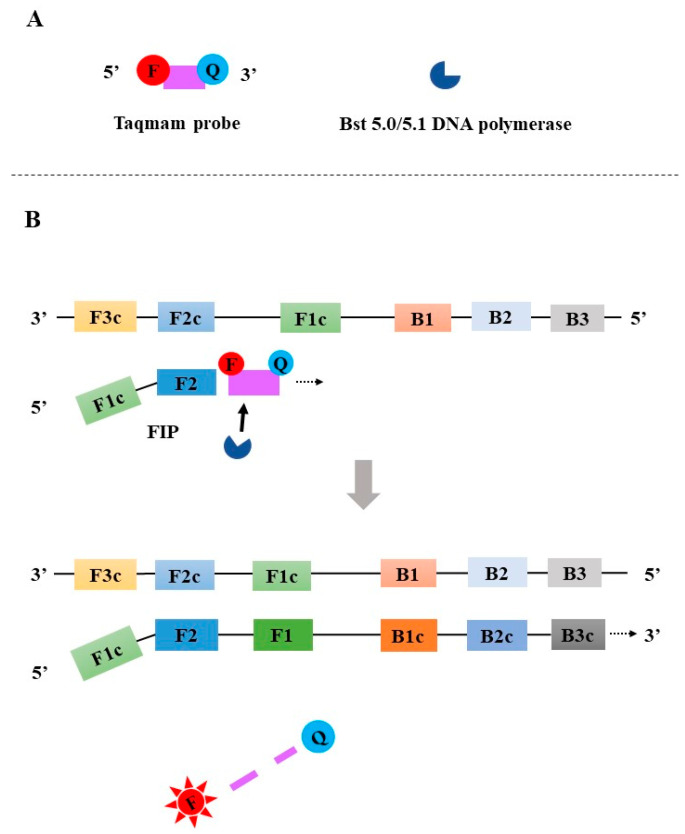
The principle of the Taqman-LAMP. F: fluorophore; Q: quencher.

**Table 2 diagnostics-13-01530-t002:** Comparison of the probe-based LAMP methods with the traditional LAMP methods.

	Traditional LAMP	Probe-Based LAMP
Reaction instrument	Thermal cycler/small portable isothermal devices/simple, low-cost device	Real-time thermal cycler/small portable isothermal fluorimeters/simple, low-cost device
Reaction time	about 60 min	less than 30 min
Detection method	Agarose gel electrophoresis/Turbidity/Fluorescence detection by non-specific fluorescent dyes or pH indicators	Real-time monitoring/end-point fluorescence detection
Specificity (non-specific signal)	Relatively low	High
Sensitivity (copies/reaction)	High	Slightly higher
Single-tube multiplex detection	No	Yes
The possibility of false positive results	Relatively high	Low
Operation	Difficult	Relatively easy
Testing cost	Slightly low	Low
Potential for commercial and/or clinical application	Low	High

**Table 3 diagnostics-13-01530-t003:** Comparison of various probe-based LAMP assays.

Methods	Probe Design	Specificity	Sensitivity (LOD, Copies per Reaction) *	One-pot Multiplexing	Variant-Tolerance	Amplification Speed (Tt Values) *
Assimilating probe	Difficult. Two partially complementary oligonucleotides need to be designed.	Relatively low. The probe is involved in one of the inner primers that more likely result in non-specific amplification, and the release of fluorescent signal depends on strand displacement of double-stranded probe.	Low (15–9.8 × 10^3^)	Yes	No	Relatively fast (4.19–27.73 min)
DARQ	Easy	Relatively low. The probe is involved in one of the inner primers that more likely result in non-specific amplification, and the release of fluorescent signal depends on strand displacement of double-stranded probe.	High (526–2477)	Yes	No	Relatively fast (11.8 ± 0.03 min)
Q Probe	Difficult. The 3′ or 5′ end of at least one loop primer is required to be cytosine for labeling a fluorescent dye.	High	High (ND)	Yes ^$^	No	Relatively slow (20.2–30.9 min)
MERT-LAMP/TEC-LAMP	Difficult. An endonuclease recognition site is required to be included in the probe.	Relatively low. The probe is involved in one of the inner primers that more likely result in non-specific amplification, and the release of fluorescent signal depends on strand displacement of double-stranded probe.	High (17.3–39.5)	Yes	No	Relatively slow (ND)
Proofman/HFman/LANTERN	Slightly easier. Any one of the loop primers can be selected as the probe by labeling a fluorophore and a quencher at both ends.	High	High (8–100)	Yes	Yes	Relatively fast (4.0–45.0 min)
TaqMan-LAMP	Difficult. The probe needs to be designed between F1 and F2 or B1 and B2.	High	High (ND)	Yes ^$^	No	Relatively fast (ND)

* Based on self-reported data. ^$^ According to the principles, it enables multiplex detection. However, there is no data to support this. “ND”, No data in the original paper.

## Data Availability

Data sharing is not applicable.

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
