# Peer review of "Evolution of the Probe-Based Loop-Mediated Isothermal Amplification (LAMP) Assays in Pathogen Detection"

_diagnostics, 2023, doi:10.3390/diagnostics13091530_

Round 1

Reviewer 1 Report

In general, this manuscript is well written and structured, and is easy to follow and understand the principles of the probe-based LAMP assays and its applications for detecting pathogens. Moreover, authors detailed each application by listing target of detection, reaction condition, sensitivity, limitation of detection, etc., and presented the comparison of various probe-based LAMP assays. Overall, authors made systematic contribution to the literature review in the field of investigating LAMP methods, and sufficient information through the manuscript can provide the guidance for the development of LAMP-based methods for researchers. Some specific comments follow:

1.       The figure 2 offers the main principles of probe-based LAMP assays. Somehow, the quality of figure is not good enough for reading some details. The font size could be also scaled up.

2.       Line 99-100 on page 4, is it clearer and better to describe “such as forward inner primers (FIP, F1c +F2), backward inner primers (BIP, B1c + B2)”? Even though the abbreviation of FIP and BIP is commonly used.

Author Response

Responses to Reviewer 1:

  1. The figure 2 offers the main principles of probe-based LAMP assays. Somehow, the quality of figure is not good enough for reading some details. The font size could be also scaled up.

Response: According to both your and the second reviewer’s suggestion, we separated figure 2 into multiple separate figures (new Figures 3-8) to exhibit the principle of each method. Accordingly, the font size was scaled up.

  1. Line 99-100 on page 4, is it clearer and better to describe “such as forward inner primers (FIP, F1c +F2), backward inner primers (BIP, B1c + B2)”? Even though the abbreviation of FIP and BIP is commonly used.

Response: Thank you for this suggestion. We have used “forward inner primer, FIP (F1c + F2) and backward inner primer, BIP (B1c + B2)” to describe them. Please see section 1, lines 50-51.

Reviewer 2 Report

The manuscript summarizes the applications of probe-based LAMP assays and discusses the advantages and shortcomings of each assay in a comprehensive and well-written manner. Given the significant impact of recent pandemics on people's lives, I believe this topic will appeal to many readers seeking reference. I recommend publication of the manuscript with major revisions.

Comments: 

1. Figure 2 contains the schemes of six different assays. The text in each assay is small and unclear. It is recommended to separate the figure into multiple figures to show the details more clearly.

2.  The manuscript compares traditional and probe-based LAMP assays, but only probe-based LAMP assays are described. To help readers unfamiliar with LAMP, I suggest adding a paragraph and a schematic illustration to show how traditional LAMP works.

3. In section 2 (the principles of the probe-based LAMP assays), some assays are not fully described, such as assimilating probe, enzyme-mediated probe-based LAMP assays, and multiple endonuclease restriction real-time LAMP. Please consider adding more illustrations to show how these assays work step-by-step.

4. Although the authors listed examples of probe-based LAMP assays in Table 1, some examples provided in Section 3 (applications of probe-based LAMP assays in pathogen detection) are outdated. Please consider replacing them with articles published after 2019.

5. In lines 229-232, the authors state that traditional LAMP has relatively low sensitivity and specificity, leading to a high potential risk of contamination. Please provide a publication or data to support this statement.

6. In line 235, the authors mention that some small portable thermostatic fluorescence analyzers have been developed. Please provide references to support this statement. Additionally, please explain why traditional LAMP assays cannot be illuminated by thermostatic fluorescence analyzers

7. In section 7 (conclusions and future perspectives), the authors summarize the manuscript but do not provide any future perspectives. I recommend the authors provide some personal opinions on how to address the shortcomings mentioned in the manuscript.

Author Response

Responses to Reviewer2:

  1. Figure 2 contains the schemes of six different assays. The text in each assay is small and unclear. It is recommended to separate the figure into multiple figures to show the details more clearly.

Response: As suggested, we separated the figure into multiple figures (new Figures 3-8), and scaled up the font size to improve the quality of each figure.

  1. The manuscript compares traditional and probe-based LAMP assays, but only probe-based LAMP assays are described. To help readers unfamiliar with LAMP, I suggest adding a paragraph and a schematic illustration to show how traditional LAMP works.

Response: As suggested, we have added a paragraph (Section 1, lines 49-63) and a schematic illustration (new Figure 1) to show how the traditional LAMP works.

  1. In section 2 (the principles of the probe-based LAMP assays), some assays are not fully described, such as assimilating probe, enzyme-mediated probe-based LAMP assays, and multiple endonuclease restriction real-time LAMP. Please consider adding more illustrations to show how these assays work step-by-step.

Response: In the revised manuscript, we present the schematics of each assay separately (new Figures 3-8) and add more detailed descriptions of how each assay works step by step (Please see revised section 2).

  1. Although the authors listed examples of probe-based LAMP assays in Table 1, some examples provided in Section 3 (applications of probe-based LAMP assays in pathogen detection) are outdated. Please consider replacing them with articles published after 2019.

Response: Thank you for this suggestion. We have replaced them with the latest articles (published after 2019), and added two newly published articles in Table 1 (Please see the updated section 3 and Table 1). Two methods (MERT-LAMP (2015) and TEC-LAMP (2018)) were not updated since there were no further literatures available since their publication.

  1. In lines 229-232, the authors state that traditional LAMP has relatively low sensitivity and specificity, leading to a high potential risk of contamination. Please provide a publication or data to support this statement.

Response: As suggested, we added necessary references to support the statement of low specificity and sensitivity of the traditional LAMP method. In fact, the major challenge to limit the wide application of the traditional LAMP method is low specificity (non-specific amplification). We did not deny that some previous papers self-reported high detection sensitivity of the traditional LAMP assays. However, for majority of them, the high sensitivity might be involved in non-specific amplification (especially when the template input was very low). This is the main reason why these reported assays were not widely and commercially used.

  1. In line 235, the authors mention that some small portable thermostatic fluorescence analyzers have been developed. Please provide references to support this statement. Additionally, please explain why traditional LAMP assays cannot be illuminated by thermostatic fluorescence analyzers.

Response: As suggested, we added a list of portable fluorescence analyzers and related refs to support the development of isothermal fluorimeters.

In fact, the traditional LAMP assays with non-specific fluorescence dyes (e.g. SYBR green I, SYTO 9) can also be illuminated by thermostatic fluorescence analyzers. However, as we mentioned above and in the main text, the major limitation of the traditional LAMP method is low specificity (non-specific amplification). Because these fluorescence dyes typically non-specifically bind to dsDNA, they are unable to distinguish specific and non-specific amplicons. The presence of non-specific amplicons will result in false-positive signal.

Distinct from the traditional LAMP method, the use of probe in LAMP assays can largely improve the specificity of the assays. In the reaction, only when the probe specifically binds the target, the fluorophore is cleaved from the probe (quencher) to release fluorescence signal, which confers the high-specificity of the probe-based LAMP assays (like probe-based qPCR assay). We added some necessary description in the revised manuscript.

  1. In section 7 (conclusions and future perspectives), the authors summarize the manuscript but do not provide any future perspectives. I recommend the authors provide some personal opinions on how to address the shortcomings mentioned in the manuscript.

Response: As suggested, we gave future perspectives and added our personal opinions.

Reviewer 3 Report

This interesting review is devoted to summarizing the achievements in the field of probe-based loop mediated isothermal amplification and its using for pathogen detection. Isothermal amplification has been a promising alternative to conventional and qPCR, being less expensive and time-consuming. I believe that the MS corresponds well to the requirements for publication in Diagnostics. At the same time, I would like to propose the authors to modify some moments to improve quality and scientific soundness of the paper.

1. First of all, I recommend the authors to read the MS text once again to eliminate some stylistic drawbacks, for example:

P. 1, lines 10-12: "the traditional LAMP..." repeats in two sebsequent sentences. I propose to unite these sentences or use "these techniques/approaches" in the second one instead of "traditional LAMP methods"

P. 1, lines 30-31: "development of PCR" repeats twice in the same sentence

P. 3, line 72-23, the sentence could be changed to (for example): "Among the methods most widely used for the detection of LAMP signal, are turbidity......."

P. 4, lines 103-105: may be "Different probes were designed for LAMP assays, including...."

Also I should note that enzymes' names should be given in italic.

2. Also I would like to propose the authors to include some additional recent papers to the reference list and discussion, such as

Gadkar et al., Sci Rep, 2018; Hyman et al, Cell Rep Met, 2022; Kline et al., Clin Microbiol, 2022; Shen et al., Front Vet Sci, 2022. Also it would make sense to include some works regarding the use of probe-based LAMP for the detection of plant pathogens.

3. It would be interesting to compare the key characteristics of probe-based LAMP not only between different types of probes, but also to conventional and/or qPCR assays (if it possible). 

At the same time I want to note that these comments are optional and the authors can decide themselves whether they have to make these corrections. 

Author Response

Responses to Reviewer3:

  1. First of all, I recommend the authors to read the MS text once again to eliminate some stylistic drawbacks, for example:
  2. 1, lines 10-12: "the traditional LAMP..." repeats in two sebsequent sentences. I propose to unite these sentences or use "these techniques/approaches" in the second one instead of "traditional LAMP methods"

Response: Thank you for this suggestion. We used "these approaches" in the second one instead of "traditional LAMP methods". Please see lines 11-12.

  1. 2, lines 30-31: "development of PCR" repeats twice in the same sentence

Response: In this sentence, we described PCR first and then quantitative (or real-time) PCR (qPCR). Considering the progressive relationship between the two, we did not revise this sentence.

  1. 3, line 72-23, the sentence could be changed to (for example): "Among the methods most widely used for the detection of LAMP signal, are turbidity......."

Response: We changed the sentence according to your suggestion. Please see section 1, lines 72-76.

  1. 4, lines 103-105: may be "Different probes were designed for LAMP assays, including...."

Response: We changed the sentence according to your suggestion. Please see section 2, line 107.

Also I should note that enzymes' names should be given in italic.

Response: The names of enzymes have been italicized. Please see section 2.4.

  1. Also I would like to propose the authors to include some additional recent papers to the reference list and discussion, such as

Gadkar et al., Sci Rep, 2018; Hyman et al, Cell Rep Med, 2022; Kline et al., Clin Microbiol, 2022; Shen et al., Front Vet Sci, 2022. Also it would make sense to include some works regarding the use of probe-based LAMP for the detection of plant pathogens.

Response: Thank you for your careful guidance. We have learned some of these references and added them to section 6 and the reference section. We expect more probe-based LAMP technologies to be developed and applied to a wider range of diagnostics.

  1. It would be interesting to compare the key characteristics of probe-based LAMP not only between different types of probes, but also to conventional and/or qPCR assays (if it possible).

Response: Thank you for your advice. As this paper focuses on the review of probe-based LAMP, we will not add more comparisons between these LAMP methods and conventional and/or qPCR assays.

At the same time I want to note that these comments are optional and the authors can decide themselves whether they have to make these corrections.

Response: Thank you for your helpful comments. We made the necessary changes mentioned above to improve our paper.

Reviewer 4 Report

In my review of the manuscript “Evolution of the probe-based loop-mediated isothermal ampli- 2 fication (LAMP) assays in pathogen detection” the author needs to correct the spelling and sentence forming errors, the topic of this manuscript is interesting and written well, just go through detailed study, follow the below comments and solve them, it will help to enrich the content of this manuscript.  Authors are suggested to refer following papers in the introduction; https://doi.org/10.3390/vaccines11020374 and 10.1016/j.crmicr.2022.100120

Minor comments:

1.       Add the reference at line 106.

2.       Correct the heading 2.4.2.

3.       If possible, not repeat the same reference again and again, the reference 27, was repeated almost 5 to 6 times.

Major comments:

1.       Add the detailed explanation of figure 6.

2.       Add the detailed explanation of figure 8

3.       Kindly explain the study limitations as well and try to highlight the novelty of work as I can see dozens of work on the same topic in the recent past

Author Response

Responses to Reviewer4:

Minor comments:

  1. Add the reference at line 106.

Response: We added the reference according to your suggestion. Please see line 107.

  1. Correct the heading 2.4.2.

Response: We corrected the heading 2.4.2. Please see lines 168-169.

  1. If possible, not repeat the same reference again and again, the reference 27, was repeated almost 5 to 6 times.

Response: Thank you for this suggestion. We have noticed the problem and reduced the number of citations to this paper.

Major comments:

  1. Add the detailed explanation of figure 6.

Response: We added the detailed explanation of figure 6. Please see section 2.4.1, lines 157-166.

  1. Add the detailed explanation of figure 8

Response: We added the detailed explanation of figure 8. Please see section 2.4.4, lines 200-206.

  1. Kindly explain the study limitations as well and try to highlight the novelty of work as I can see dozens of work on the same topic in the recent past years.

Response: We don't deny that the study has some limitations. As we describe in Table 3, some probe-based LAMP methods have difficulties at probe design, and several factors can affect the specificity of the detection. In addition, underlying questions in result reading and nucleic acid extraction remain to be further addressed in the probe-based LAMP methods (Section 6). In recent years, LAMP assays for various pathogens were developed by using different probe-based LAMP methods, which are ascribed to the certain advantages of these methods (e.g. comparable sensitivity and specificity with qPCR technique, rapid testing time, low cost, results can be read in a variety of ways, flexibility in combination with other technologies, etc.) and the prospect of commercial detection. Given the huge demand in the pathogen testing market, we believe the topic of this review will attract researchers to further develop probe-based LAMP approaches, facilitating their commercialization and clinical diagnosis application.

Round 2

Reviewer 2 Report

The authors have addressed every comments. Acceptance is suggested. 

Reviewer 4 Report

I have no further comments